# Fear among Syrians: A Proposed Cutoff Score for the Arabic Fear of COVID-19 Scale

**Fatema Mohsen**[1☉]*, **Batoul Bakkar**[1☉], **Salma Khadem alsrouji**[1☉], **Esraa Abbas**[1☉], **Alma Najjar**[1☉], **Marah Marrawi**[2☉], **Youssef Latifeh**[3,4☉]

**1** Faculty of Medicine, Syrian Private University, Damascus, Syria, **2** Department of Statistics, Syrian Private University, Damascus, Syria, **3** Faculty of Medicine, Department of Psychiatry, Syrian Private University, Damascus, Syria, **4** Faculty of Medicine, Department of Psychiatry, Damascus University, Damascus, Syria

☉ These authors contributed equally to this work.
* fatemamohsena@gmail.com

## Abstract

COVID-19 pandemic has led to psychological health issues one of which is fear. This study validates the Arabic version of the fear of COVID-19 scale and suggests a new cutoff score to measure fear of COVID-19 among the Syrian Population. A total of 3989 participants filled an online survey consisting of socio-demographic information, the fear of COVID-19 scale, the patient health questionnaire 9-item, and the generalized anxiety disorder 7-item. Receiver operating characteristic analysis was used to define cutoff scores for the fear of COVID-19 scale in relation to generalized anxiety disorder 7-item and the patient health questionnaire 9-item. The Cronbach α value of the Arabic fear of COVID-19 scale was 0.896, revealing good stability and internal consistency. The inter-item correlations were between [0.420–0.868] and the corrected item-total correlations were between [0.614–0.768]. A cutoff point of 17.5 was deduced from the analysis. According to the deduced cutoff point, 2111(52.9%) were categorized as extreme fear cases. This cutoff score deduced from this study can be used for screening purposes to distinguish community members that may be prone to developing extreme fear of COVID-19. Therefore, early preventive and supportive measures can then be delivered.

## Introduction

The novel Coronavirus disease-2019 (COVID-19) is a momentous danger to humanity's health that has emerged as an outbreak. The first COVID-19 incident was noted in Wuhan, China in December 2019. This virus quickly crossed the borders, spreading all over the world, resulting in a never-ending pandemic burdened with the results of morbidity and mortality [1]. There are live updates of the latest cases, and deaths every second of the day, as well as official briefings, and prognostications about future cataclysmic events [2]. Enumeration of all deaths, compared to the 0.28% crude mortality rate, which will continue to rise as more infections and deaths occur, produces an image of excess death, capturing both the direct burden of the pandemic and its indirect mortality burden [3,4]. 22 March 2020 marked the first officially reported COVID-19 case in Syria [5]. The numbers have escalated since then with Syria now entering its third wave [6–8].

**Data Availability Statement:** All relevant data are within the paper and its files.

**Funding:** The author(s) received no specific funding for this work.

**Competing interests:** The authors have declared that no competing interests exist.

**Abbreviations:** COVID-19, Coronavirus Disease 2019; WHO, World Health Organization; PHEIC, Public Health Emergency of International Concern; FCV-19S, Fear of COVID-19 Scale; PHQ-9, Patient Health Questionnaire 9-item; GAD-7, Generalized Anxiety Disorder 7-item; ROC, Receiver operating characteristic; IRB, Institutional Review Board; SPSS, Statistical Package for Social Sciences; AVE, Average Variance Extracted; CR, Composite Reliability; HTMT, Heterotrait-Monotrait; SD, Standard Deviation; AUC, Area Under the Curve.

As the pandemic has embedded misconceptions around COVID-19, strict precautionary measures have been adopted by governments, such as physical distancing, self-isolation, and quarantine [9]. These measures were implemented to curtail the tide of the pandemic when no treatments or vaccines were available during the time of survey administration. During the Syrian lockdown, many streets have become ghost towns. Flights all over the globe have ceased. Business and medical conferences have been postponed to the unknown future. Mosques, museums, parks, schools, and universities have been closed and abandoned. Grocery and pharmacy stores are being cleared of over-the-counter medications, disinfectants, supplements, and herbal remedies. Households are stacked to the brim with groceries and toiletries, in preparation for a long draconian lockdown that may never end. National closures of businesses, public facilities, and entertainment have resulted in less currency in circulation. All the above have shocked the economy drastically. The effects of the mass media and governments have had a toll on society, probably for several years to come; their disruptive actions are eliciting more damage to people around the globe than COVID-19. As a major health issue, the COVID-19 pandemic has triggered a multilevel global crisis, affecting an individual's physical and psychological mental health. Thus, researchers are expressing concerns related to COVID-19 adverse effects on society's mental health and psychological well-being [10–22].

One of these psychological impacts is fear, a vital emotion necessary for an individual's survival by arousing adaptive defense responses against potentially threatening events or danger [23]. The current threatening event is COVID-19, a prolonged worry or negative emotional reaction towards contracting COVID-19, morbidity, and mortality [24]. Added to the previous are the effects of the pandemic on our daily habits including unemployment, working from home, home-schooling, and lack of socializing. Fear is becoming a critical concern, as the pandemic continues to spread high infection, morbidity, and mortality cases. These concerns have pushed researchers to assess the fear of COVID-19 in different regions of the world and identify the possible triggers [25].

Fear may contribute to the development of mental health disorders or further deteriorate pre-existing psychiatric symptoms such as generalized anxiety disorders, depression, and post-traumatic stress disorder, and even suicide [12,26,27]. A recent study has developed the Fear of COVID-19 Scale (FCV-19S) as a means of boosting the medical practice that is dedicated to preventing the spread of COVID-19 cases and treating them [28]. The psychometric evidence of the FCV-19S demonstrated satisfactory results [29–35]. A few studies thereafter have validated the questionnaire in various languages and examined associations with socio-demographic characteristics [33,36–42]. However, only one study as of yet has proposed a cutoff score for the FCV-19S [43]. Given that Syria is one of the most vulnerable countries in the world to be tormented by mental health disorders due to both COVID-19 and war. The aim of the study is to assess the fear regarding COVID-19 among Syrians, validate the FCV-19S, and identify an appropriate cutoff score to differentiate individuals with normal fear of COVID-19 response from ones encountering extreme fear.

## Materials and methods

### Study design, setting, and participants

A web-based cross-sectional study was conducted using an Arabic questionnaire over a period of 12 days between May 2 and May 14 of 2020. Inclusion criteria: Participants aged 18 and above, residents in Syria, and full completion of the survey. A convenience sampling method was used in the study. The questionnaire was uploaded over several popular social media apps. Once informed consent was received, participants filled in their socio-demographic information including sex, age, place of residence, education level, occupation, and economic status.

The socio-demographic questions in Arabic and English are displayed in S1 and S2 Tables. Participants were also asked about the history of chronic diseases. The total dataset of 3989 participants was used to validate the Arabic version of the FCV-19S.

## Development of the arabic FCV-19S

The English FCV-19S was translated into Arabic through a forward-backward translation technique [28]. The FCV-19S was translated into Arabic by two medical translators who are fluent in both Arabic and English. The authors, who are fluent in Arabic and English, revised the Arabic translation and no errors were identified with the translation. The final copy of the Arabic translation was back-translated to English by another medical translator who was unacquainted with the original English FCV-19S. A comparison of the forward and backward scale translations to identify any differences and cultural convenience was done. A pilot study including 20 volunteers was undertaken to assess the scale's reliability, clarity, relevance, and acceptability of the survey, who were eliminated from the statistical analysis to avoid bias. No modifications were required. The Arabic version of the FCV-19S is provided in S3 Table.

## Measures

The FCV-19S (after translation) was used to evaluate the symptoms of fear. The FCV-19S is a one-dimensional scale that measures one's fear level of COVID-19 it consists of 7 items. Each item is rated on a 5-point Likert scale with 1 representing strongly disagree and 5 representing strongly agree, providing a 7 to 35 total score range. The higher the score the greater the levels of fear of COVID-19 are [41].

The Arabic Patient Health Questionnaire 9-item (PHQ-9) was used in the study to assess depression symptom severity [44]. Items on the PHQ-9 were scored over a 4-point Likert scale: 0 = not at all, 1 = several days, 2 = more than half the days, and 3 = nearly every day), providing a 0–27 severity score range. The scores were categorized into 5 groups: none (0–4), mild (5–9), moderate (10–14), moderately severe (15–19), and severe (20–27).

The Arabic Generalized Anxiety Disorder 7-item (GAD-7) was used in this study to assess anxiety symptom severity [45]. Items on the GAD-7 were scored over a 4-point Likert scale: 0 = not at all, 1 = several days, 2 = more than half the days, 3 = nearly every day, providing a 0–21 severity score range. The Gad-7 is a self-rated scale used to evaluate the severity of the 4 most common anxiety disorders (Generalized Anxiety Disorder, Panic Disorder, Social Phobia, and Post Traumatic Stress Disorder), a cutoff score of 0–4 indicates no anxiety symptoms. The scores were categorized into 4 groups: none (0–4), mild (5–9), moderate (10–14), and severe (15–21) [46].

## Ethics

Ethical approval was granted by the Institutional Review Board (IRB) of the Faculty of Medicine, Syrian Private University. The study was not granted a registered number.

## Statistical analysis

Descriptive analysis, including frequencies, percentages, means, and standard deviations (SD) were applied. The Cronbach's α test and inter-item correlation were used to assess the internal consistency, with satisfactory reliability set at $\geq 0.70$ and between 0.20 and 0.40 respectively [47,48]. The corrected item-total correlation was used to assess the coherence of the FCV-19S (values > 0.4 are acceptable). The item factor loadings, average variance extracted (AVE), and composite reliability (CR) were used to assess the convergent validity of the FCV-19S. The

Heterotrait-Monotrait (HTMT) was used to assess the discriminant validity of the FCV-19S. A CR value between 0.7 and 0.9, AVE value > 0.5, and discriminant validity < 0.85 were set for this study [49–51]. Cutoff scores were deduced from receiver operating characteristic (ROC) analysis for the FCV-19S scale in association with GAD-7 and PHQ-9 (external criteria). The Youden-Index was used to deduce the optimal cutoff score for the Arabic scale and to lower the risk of miscategorization. A dichotomous variable was created out of the total for both GAD-7 and PHQ-9 using the cutoff point of 10 to assess anxiety and depressive symptoms respectively [44,46]. After determining the cutoff points, participants with a total score above the deduced cutoff value were categorized as cases with extreme fear, while those below it were categorized under normal fear of COVID-19. The Statistical Package for Social Sciences version 25.0 (SPSS Inc., Chicago, IL, United States) was used.

## Results

### Socio-demographic characteristics of participants

Of 5000 total participants invited to take part in the study, 4,430 gave informed consent. A final sample size of 3989 participants (response rate = 79.8%) met the inclusion criteria for the study. The 441 participants who did not meet the inclusion criteria were either below 18, residing outside Syria, or uncompleted survey. Most participants were female 2935 (73.5%), single 3096 (77.6%), students 2397 (60.1%), and residing in Damascus 1412 (35.4%). Ages ranged between 18 and 70 years. The major age group was 18–25 years 2870 (71.9%). A total of 416 (10.4%) and 1522 (38.1%) participants reported their economic status as under poor and moderate respectively. 556 (15.9%) mentioned a history of chronic diseases Table 1.

### Validity and cutoff Score analysis

The FCV-19S scores ranged between 7 and 35, the mode score was 14, and the mean score was 18.5 (± 6.009) Fig 1. Participants' responses to the FCV-19S are shown in Table 2. The Cronbach α value of the Arabic FCV-19S was 0.896, revealing good stability and internal consistency. The inter-item correlations were between [0.420–0.868] and the corrected item-total correlations were between [0.614–0.768] Table 3. The CR value for all items of the FCV-19S was 0.934, exceeding the set threshold, deeming the CR unreliable. The AVE value for all items of the FCV-19S was good (0.672). The value 0.820 was a measure of reasonable discriminant validity of the FCV-19S.

The responses to the GAD-7 scale were categorized into two groups: non-anxious symptoms (score < 9) and anxious symptoms (9 < score ≤ 21). The responses to the PHQ-9 scale were categorized into two groups: non-depressed symptoms (score < 9) and depressed symptoms (9 < score ≤ 27) Table 4.

ROC analysis was used to evaluate the efficacy of the FCV-19S in predicting anxiety and depression factors. The proposed cutoff score was determined by the optimal sensitivity and specificity level, Table 5, revealing the following

- Unsatisfactory predictive power for the FCV-19S in disclosure of anxiety symptoms assessed using GAD-7 was found. The best cutoff was at 18.5, with a sensitivity of 53% and specificity of 57.5%, as the area under the curve (AUC) was 0.560 with a statistical significance (p-value < 0.001), a confidence interval of 95% CI = [0.541, 0.578] Table 5 and Fig 2.

- Unsatisfactory predictive power for the FCV-19S in disclosure of depressive symptoms assessed using PHQ-9 was found. The best cutoff was at 17.5 with a sensitivity of 55.1% and

**Table 1. Participants socio-demographics.**

| | | |
|---|---|---|
| **Gender (%)** | **Male** | 1054 (26.4) |
| | **Female** | 2935 (73.5) |
| **Age (%)** | **18–25** | 2870 (71.9) |
| | **26–34** | 685 (17.2) |
| | **35–44** | 261(6.5) |
| | **45–54** | 121 (3.0) |
| | **55 <** | 52 (1.4) |
| **Social Status (%)** | **Single** | 3096 (77.6) |
| | **Married** | 714 (17.9) |
| | **Other** | 179 (4.5) |
| **Economic Status (%)** | [1]**Excellent** | 251 (6.3) |
| | [2]**Good** | 1800 (45.1) |
| | [3]**Moderate** | 1522 (38.1) |
| | [4]**Poor** | 416 (10.4) |
| **Chronic disease(s) (%)** | **Yes** | 556 (13.9) |
| | **No** | 3433 (86) |
| **Education (%)** | **Primary School** | 21 (0.5) |
| | **Intermediate School** | 115 (2.9) |
| | **Secondary school** | 370 (9.3) |
| | **College/ University** | 3271 (82) |
| | **Master's degree** | 185 (4.6) |
| | **PhD** | 27 (0.7) |
| **Occupation (%)** | **Health care worker** | 259 (6.5) |
| | **Government institution** | 239 (6.0) |
| | **Private institution** | 202 (5.1) |
| | **Business** | 202 (5.1) |
| | **Military** | 35 (0.9) |
| | **Student** | 2397 (60.1) |
| | **Other** | 655 (16.4) |
| **Household members (%)** | **Alone** | 54 (1.4) |
| | **1–5** | 2474 (62) |
| | **>5** | 1461 (36.6) |

[1]Poor: Income does not provide essential needs for the family.

[1]Excellent: Income provides luxury requirements.

[2]Good: Income provides essential needs and some luxury requirements.

[3]Moderate: Income provides essential needs for the family but no more.

[4]Poor: Income does not provide essential needs for the family.

specificity of 49.6%, AUC = 0.520 was statistically significant (p-value < 0.001), a confidence interval of 95% CI = [0.502, 0.538] Table 5 and Fig 3.

A cutoff point of 17.5 was deduced from the anxiety and depression ROC analysis. According to the deduced cutoff point, the majority 2111(52.9%) were categorized as cases with extreme fear Table 4.

## Discussion

The current study was performed to validate the Arabic version of the FCV-19S and propose a cutoff to distinguish members of the population who reacted towards COVID-19 with extreme

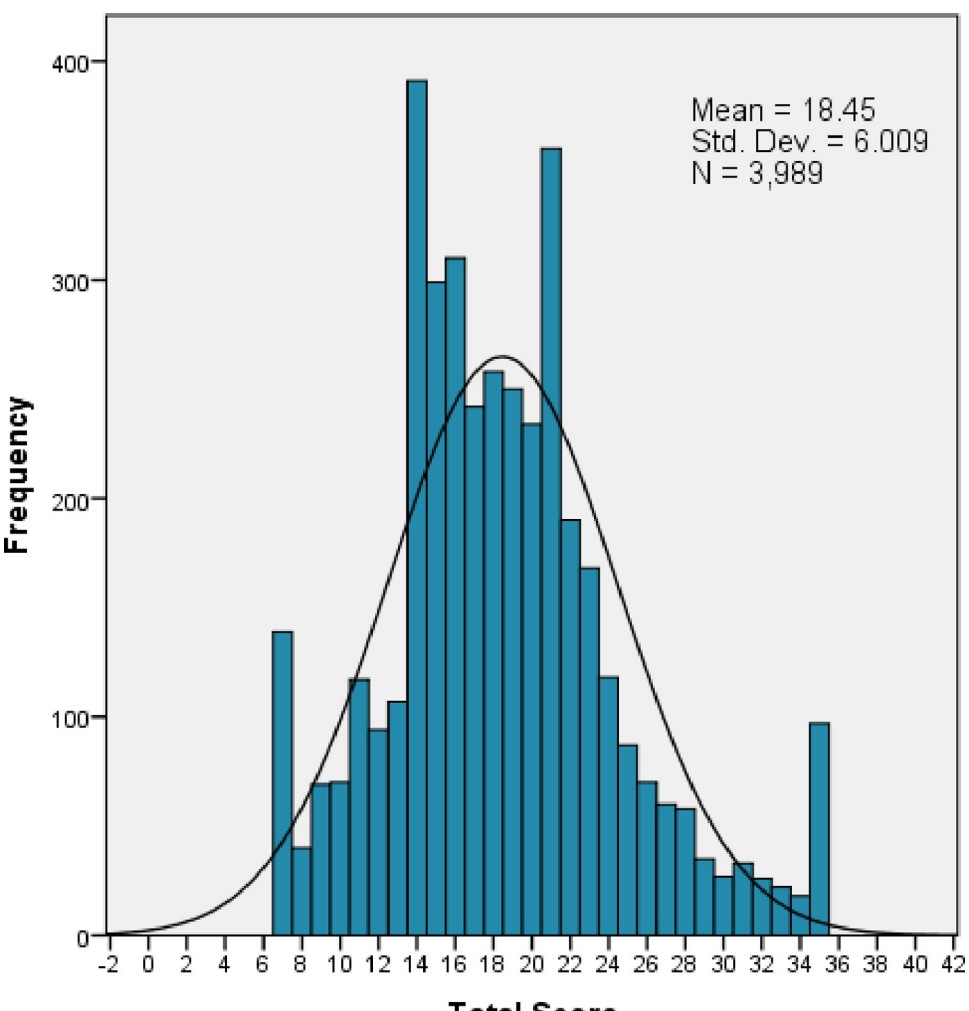

**Fig 1. Participant's FCV-19S score distribution.**

**Table 2. Participants response to FCV-19S.**

| FCV-19S | Response to FCV-19S | | | | |
|---|---|---|---|---|---|
| | Totally disagree (%) | Disagree (%) | Neutral (%) | Agree (%) | Totally Agree (%) |
| 1- I am often afraid of the emerging COVID-19 | 261 (6.5) | 826 (20.7) | 1557 (39.0) | 951 (23.8) | 394 (9.9) |
| 2- I feel uncomfortable when I think about the new COVID-19 | 279 (7.0) | 906 (22.7) | 1115 (28.0) | 1228 (30.8) | 461 (11.6) |
| 3- My hands get sweaty when I think about the new COVID-19 | 962 (24.1) | 1983 (49.7) | 719 (18.0) | 119 (3.0) | 206 (5.2) |
| 4- I am afraid of dying due to infection with the emerging COVID-19 | 685 (17.2) | 1579 (39.6) | 892 (22.4) | 463 (11.6) | 370 (9.3) |
| 5- I feel anxious and nervous when I follow news or posts about COVID-19 on social media | 390 (9.8) | 968 (24.3) | 1002 (25.1) | 1070 (26.8) | 559 (14.0) |
| 6- I cannot sleep because of my anxiety about infection with the emerging COVID-19 | 1045 (26.2) | 1875 (47.0) | 688 (17.2) | 152 (3.8) | 229 (5.7) |
| 7- I feel my heart racing and palpitations when I think about the emerging COVID-19 | 1023 (25.6) | 1843 (46.2) | 670 (16.8) | 211 (5.3) | 242 (6.1) |

**Table 3. FCV-19S construct validity analysis.**

| Items | Factor loading | Corrected item-total correlation | Cronbach's alpha | Inter-item correlation range | M(SD) | Skewness | kurtosis | Kaiser-Meyer-Olkin |
|---|---|---|---|---|---|---|---|---|
| 1- I am often afraid of the emerging COVID-19 | 0.843 | 0.661 | 0.896 | 0.423 - 0.717 | 3.1 (1) | -0.021 | -0.470 | 0.864 |
| 2- I feel uncomfortable when I think about the new COVID-19 | 0.877 | 0.655 | | 0.424–0.717 | 3.17 (1.1) | -0.146 | -0.801 | |
| 3- My hands get sweaty when I think about the new COVID-19 | 0.877 | 0.723 | | 0.420–0.791 | 2.15 (1) | 1.147 | 1.418 | |
| 4- I am afraid of dying due to infection with the emerging COVID-19 | 0.551 | 0.724 | | 0.513–0.612 | 2.576 (1.2) | 0.607 | -0.450 | |
| 5- I feel anxious and nervous when I follow news or posts about COVID-19 on social media | 0.715 | 0.614 | | 0.420–0.574 | 3.11 (1.2) | -0.067 | -0.973 | |
| 6- I cannot sleep because of my anxiety about infection with the emerging COVID-19 | 0.905 | 0.763 | | 0.429–0.868 | 2.16 (1) | 1.114 | 1.077 | |
| 7- I feel my heart racing and palpitations when I think about the emerging COVID-19 | 0.889 | 0.768 | | 0.453–0.868 | 2.2 (1.1) | 1.050 | 0.749 | |

fear from those with a normal fear reaction. Deducing a cutoff is a pivotal and familiar practice among the field of psychiatry to differentiate those into either morbid cases or nonmorbid cases [52–56]. Identifying cases and non-cases with extreme fear will allow us to stratify those in need for further assessment by a specialist in psychiatry. Once assessed, appropriate management can be given if indicated. Although many studies have used the FCV-19S to assess fear among various populations, without a cutoff score we cannot further analyze the results and compare and interpret findings with different populations [57].

Our study revealed a good internal consistency (Cronbach's α = 0.896). Our finding was in line with other studies with a range of 0.82 to 0.88 [28,42,58,59]. The inter-item correlations were between [0.420–0.868] and the corrected item-total correlations were between [0.614–0.768], whereas it was [0.35–0.66] and [0.57–0.74] respectively in a Saudi-Arabian study [60].

ROC curves were constructed to determine the FCV-19S cutoff point. A cutoff point of 17.5 was deduced, participants scoring ≥ 17.5 were categorized as having extreme fear of COVID-19, whereas participants scoring below this threshold were categorized as having normal fear of COVID-19. Only one study has defined a cutoff for the FCV-19S, with a threshold of 16.5 and higher [43].

This study revealed a weak discriminatory ability from the AUC outcomes, whereas its accuracy was moderate in a Greek study [61]. The low precision of AUC signifies that not all individuals with depression or anxiety symptoms have an unhealthy fear of COVID-19 and vice versa. This may be attributed to over a decade of conflict and the implications it induced on the mental health of the Syrian population [62]. Thus, Syrians have countless factors contributing to their depressive and anxiety symptoms unlike the majority of the globe. Also, the FCV-19S was designed to assess fear towards COVID-19 while the GAD-7 and PHQ-9 are not aimed at assessing depression and anxiety towards the COVID-19 pandemic.

**Table 4. Classification of participants according to FCV-19S, GAD-7, and PHQ-9 cutoff points.**

| | Yes (%) | No (%) |
|---|---|---|
| **FCV-19S** | 211 (52.9) | 1878 (47.1) |
| **GAD-7** | 1484 (37.2) | 2505 (62.8) |
| **PHQ-9** | 2176 (54.6) | 1813 (45.4) |

**Table 5. Prediction validity of the FCV-19S.**

|  | FCV-19S Score | Sensitivity | Specificity | AUC | CI (95%) | p-value | Youden's Index |
|---|---|---|---|---|---|---|---|
| **Anxiety** | 18.5 | 0.532 | 0.575 | 0.560 | [0.541, 0.578] | < 0.001 | 0.107 |
| **Depression** | 17.5 | 0.551 | 0.496 | 0.520 | [0.502, 0.538] | < 0.001 | 0.047 |

Fear of COVID-19 may be a factor that impacts an individual's protective behaviors, including vaccination uptake [63]. Therefore, one should have a certain level of fear but not extreme fear as individuals lack the perception to respond to threats faced [64]. Our study revealed a prevalence of 52.9% of Syrians was categorized as having extreme fear towards COVID-19. The prevalence of extreme fear among Syrians was higher in comparison with the Greeks (40.3%), despite their lower cutoff. The higher prevalence among Syrians can be attributed to the devastating impacts of war, driving the country into a prolonged economic recession. The devastating conflict has resulted in a Syrian healthcare system that is drastically unequipped to restrain such a pandemic. With many healthcare professionals fleeing the country, a low number of ventilators, and many hospitals bombarded to rubble, Syrians must feel constant fear for their lives [65].

All the above factors have exacted immense effects on the mental health of Syrians. The Syrian ministry of health provides 3 hospitals for substance abuse and mental illness: Ibn Rushd Hospital in Damascus, Ibn Sina Hospital in Rural Damascus, and Ibn Khaldoun Hospital in Aleppo. Ibn Khaldoun Hospital was bombarded on 25/12/2012, leaving only 2 working hospitals. Our study showed that 54.6% and 37.2% of Syrians are suffering from moderate to severe depressive and anxiety symptoms respectively. Grievously, 90% of cases are left unattended,

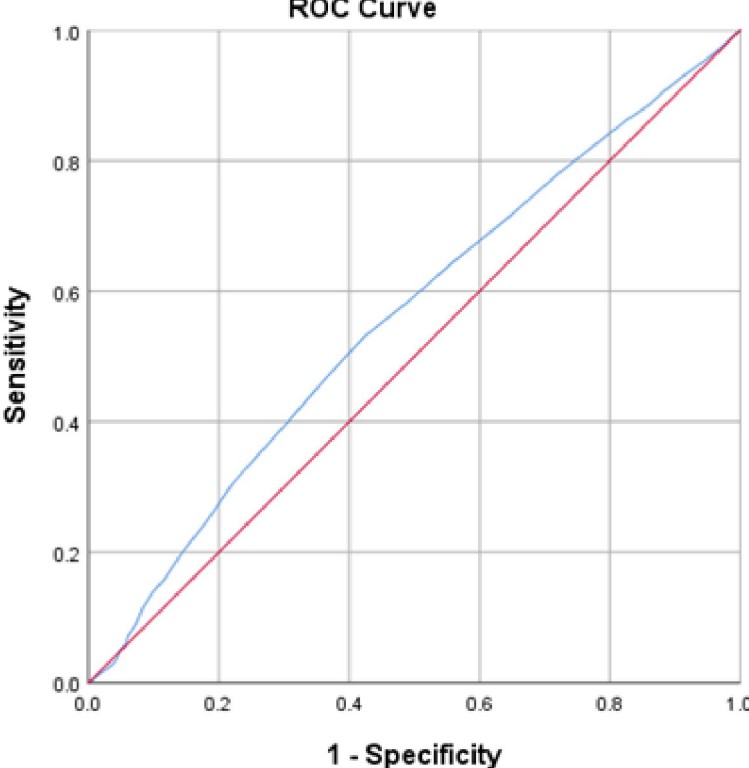

**Fig 2. ROC curve of FCV-19S for anxiety.**

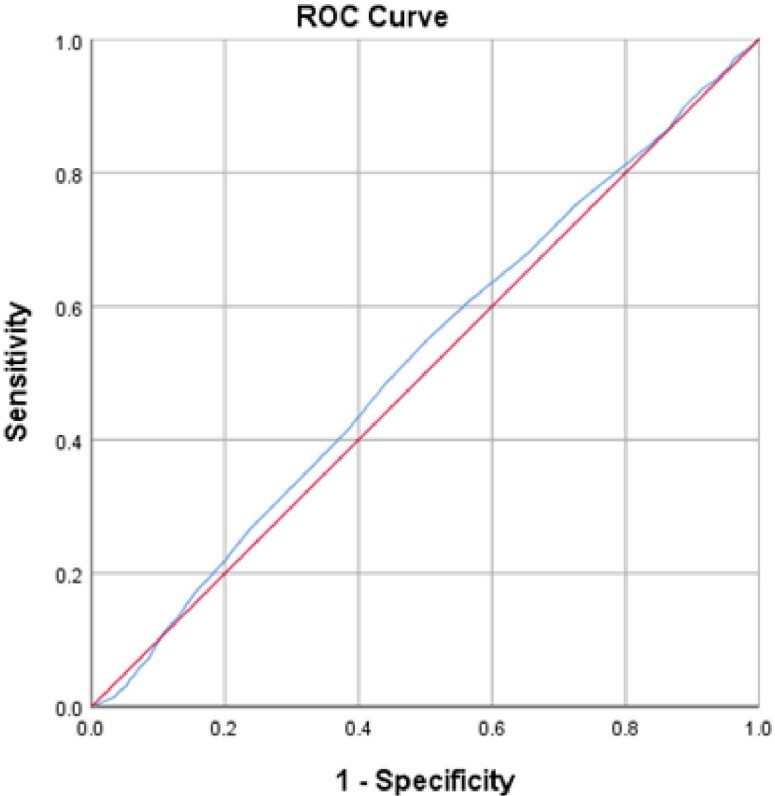

**Fig 3. ROC curve of FCV-19S for depression.**

untreated, and unmanaged, as only 80 psychiatrists are available in Syrian territories [66,67]. We must call on national and international organizations to aid us Syrians, reconstruct mental health services, and assist in providing skilled healthcare workers for the suffering people of Syria.

## Limitations

The existing study has countless limitations in its build. First, the sampling approach, convenience, may have affected the generalizability of the findings. Less educated and socially underprivileged participants were underrepresented. Second, to evaluate the representativeness of our sample, credible national Syrian data regarding demographic features of the population is required, which are currently not available. Third, the cross-sectional design limited the assessment of prospective effects of fear over time. Fourth, the proposed cutoff aimed to further make use of the scale to screen for cases with extreme fear but not for diagnostic purposes. Fifth, insufficiency of other scales assessing the same construct during the conduct of the study such as health anxiety and posttraumatic stress disorder. Sixth, the cutoff should ideally only be used on Syrian populations due to the characteristics of the participants in the study. Therefore, the proposed cutoff score must be used with attentiveness. Seventh, given the ongoing turmoil in Syria, the effects of wartime crisis perceptions concerning the COVID-19 pandemic adverse health, economic, and social ramifications on everyday life warrants further assessment.

## Conclusion

This study assessed COVID-19-related fear among the Syrian population and determined a cutoff score of $\geq 17.5$ with unsatisfactory predictive power for depression and anxiety. This

cutoff score can be used for screening purposes to identify individuals most vulnerable to developing fear-related symptomatology. Therefore, require further assessment to identify high-risk individuals and deliver early preventive and/or supportive measures. However, the Syrian healthcare system is under constant strain due to the burdens of war and the COVID-19 pandemic, and such measures may only exist in our imaginations.

## Supporting information

**S1 Table. English-version of the socio-demographic characteristics questions.**
(DOCX)

**S2 Table. Arabic-version of the socio-demographic characteristics questions.**
(DOCX)

**S3 Table. Arabic-version of FCV-19S.**
(DOCX)

## Acknowledgments

We thank the Syrian Private University for the support provided. We would like to also thank the participants in the study.

## Author Contributions

**Conceptualization:** Fatema Mohsen, Batoul Bakkar.

**Formal analysis:** Fatema Mohsen, Salma Khadem alsrouji, Marah Marrawi.

**Methodology:** Batoul Bakkar.

**Writing – original draft:** Fatema Mohsen, Salma Khadem alsrouji, Esraa Abbas, Alma Najjar.

**Writing – review & editing:** Fatema Mohsen, Youssef Latifeh.

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
