## [Decision Letter · Decision Letter 0]

25 May 2021

PONE-D-21-14780

Fear among Syrians: a Proposed Cutoff Score and Validation of the Arabic Fear of COVID-19 Scale- A National Survey

PLOS ONE

Dear Dr. Mohsen,

Thank you for submitting your manuscript to PLOS ONE. After careful consideration, we feel that it has merit but does not fully meet PLOS ONE’s publication criteria as it currently stands. Therefore, we invite you to submit a revised version of the manuscript that addresses the points raised during the review process.

We look forward to receiving your revised manuscript.

Kind regards,

Ali B. Mahmoud, Ph.D.

Academic Editor

PLOS ONE

Journal Requirements:

2. We note that your Supporting Information file may include questionnaire items that may have been previously published. The reproduction of previously published work has implications for the copyright that may apply to these publications. We would be grateful if you could clarify whether you have obtained permission from the original copyright holder to republish these items under a CC BY license. If you have not obtained permission to publish these items please remove them from your manuscript. You may wish to replace the text you have removed with relevant question numbers/ brief descriptions of each item; please be sure to include any relevant references and in-text citations.

3. Please amend the manuscript submission data (via Edit Submission) to include author Salma Khadem alsrouji.

5.Thank you for submitting the above manuscript to PLOS ONE. During our internal evaluation of the manuscript, we found significant text overlap between your submission and the following previously published works.

- https://www.statnews.com/2020/03/16/coronavirus-serious-threat-prepare-not-overreact/

We would like to make you aware that copying extracts from previous publications, especially outside the methods section, word-for-word is unacceptable, even for works which you authored. In addition, the reproduction of text from published reports has implications for the copyright that may apply to the publications.

Please revise the manuscript to rephrase the duplicated text, cite your sources, and provide details as to how the current manuscript advances on previous work. Please note that further consideration is dependent on the submission of a manuscript that addresses these concerns about the overlap in text with published work.

Reviewers' comments:

Reviewer's Responses to Questions

**Comments to the Author**

1. Is the manuscript technically sound, and do the data support the conclusions?

Reviewer #1: Partly

Reviewer #2: Partly

2. Has the statistical analysis been performed appropriately and rigorously? 

Reviewer #1: Yes

Reviewer #2: Yes

3. Have the authors made all data underlying the findings in their manuscript fully available?

Reviewer #1: Yes

Reviewer #2: Yes

4. Is the manuscript presented in an intelligible fashion and written in standard English?

Reviewer #1: Yes

Reviewer #2: Yes

5. Review Comments to the Author

Reviewer #1: This is a well reported study. However, the following points need clarification:

1) The authors state that sample size was calculated for a population of 17, 500, 657 people (line 100). It is well known that much less Syrians were living in Syria in 2020.

2) The authors imply that they tested a sample that is representing all Syrians. It is not clear how the 5000 participants were chosen to be invited to take part in the study.

3) This is web-based cross-sectional study employing social media. It is unlikely that all the people appearing in a random sample would have an online access to social media.

4) There is a contradiction between the statement (The original FCV-19S was translated into Arabic using a forward-backward translation technique – line 110) and the statement (For this study, we used the Arabic version of FCV-19S, which was validated in a previous study – line 123).

5) The title “Fear among Syrians: a Proposed Cutoff Score and Validation of the Arabic Fear of COVID-19 Scale- A National Survey” implies that that the results of this survey could generalised to the whole Syrian nation. This does not seem to be the case as the authors acknowledge that They has used “convenience sampling” – line 244, and the results cannot be generalized. This study, thus, cannot be described as a national survey in the title and in the text.

6) the emphasis in the title on a “proposed cut-off point” that is useless is not appropriate.

Reviewer #2: I believe that the present study can add merits to the present understanding of fear toward COVID-19. To the best of my knowledge, although many studies have translated and validated the Fear of COVID-19 Scale (FCV-19S), none have proposed a cutoff for the FCV-19S. Therefore, the major contribution of the present study is the proposed cutoff with using previous standardized fear scales. Moreover, the present study has the strengths of (1) rigorous translation for the FCV-19S, (2) large sample size of 3989, and (3) adequate statistical methods. Therefore, I read this paper with greatest interests. However, several points should be addressed before I recommend publication. Please see my comments below.

1. I would like the authors to provide more information regarding the psychometric evidence of the FCV-19S. The following references are recommended to be included in a revision.

Chang, K.-C., Hou, W.-L., Pakpour, A. H., Lin, C.-Y., Griffiths, M. D. (accepted). Psychometric testing of three COVID-19-related scales among people with mental illness. International Journal of Mental Health and Addiction.

Pakpour, A. H., Griffiths, M. D., Chang, K.-C., Chen, Y.-P., Kuo, Y.-J., & Lin, C.-Y. (2020). Assessing the fear of COVID-19 among different populations: A response to Ransing et al. (2020). Brain, Behavior, and Immunity, 89, 524-525.

Andrade, E. F., Pereira, L. J., Oliveira, A., Orlando, D. R., Alves, D., Guilarducci, J. S., & Castelo, P. M. (2020). Perceived fear of COVID-19 infection according to sex, age and occupational risk using the Brazilian version of the Fear of COVID-19 Scale. Death studies, 1–10. Advance online publication. https://doi.org/10.1080/07481187.2020.1809786

Haktanir, A., Seki, T., & Dilmaç, B. (2020). Adaptation and evaluation of Turkish version of the fear of COVID-19 Scale. Death studies, 1–9. Advance online publication. https://doi.org/10.1080/07481187.2020.1773026

Caycho-Rodríguez, T., Valencia, P. D., Vilca, L. W., Cervigni, M., Gallegos, M., Martino, P., Barés, I., Calandra, M., Rey Anacona, C. A., López-Calle, C., Moreta-Herrera, R., Chacón-Andrade, E. R., Lobos-Rivera, M. E., Del Carpio, P., Quintero, Y., Robles, E., Panza Lombardo, M., Gamarra Recalde, O., Buschiazzo Figares, A., White, M., … Burgos Videla, C. (2021). Cross-cultural measurement invariance of the fear of COVID-19 scale in seven Latin American countries. Death studies, 1–15. Advance online publication. https://doi.org/10.1080/07481187.2021.1879318

Soares, F. R., Afonso, R. M., Martins, A. P., Pakpour, A. H., & Rosa, C. P. (2021). The fear of the COVID-19 Scale: validation in the Portuguese general population. Death studies, 1–7. Advance online publication. https://doi.org/10.1080/07481187.2021.1889722

Moreta-Herrera, R., López-Calle, C., Caycho-Rodríguez, T., Cabezas Guerra, C., Gallegos, M., Cervigni, M., Martino, P., Barés, I., & Calandra, M. (2021). Is it possible to find a bifactor structure in the Fear of COVID-19 Scale (FCV-19S)? Psychometric evidence in an Ecuadorian sample. Death studies, 1–11. Advance online publication. https://doi.org/10.1080/07481187.2021.1914240

2. In the second paragraph of the Introduction, the authors should include the following references, which also support the statement that COVID-19 impacts on individuals' physical and mental health.

Fung, X. C. C.#, Siu, A., Potenza, M. N., O'Brien, K. S., Latner, J. D., Chen, C.-Y., Chen, I.-H., & Lin, C.-Y. (2021). Problematic use of internet-related activities and perceived weight stigma in schoolchildren: A longitudinal study across different epidemic periods of COVID-19 in China. Frontiers in Psychiatry.

Malik, S., Ullah, I., Irfan, M., Ahorsu, D. K., Lin, C.-Y., Pakpour, A. H., Griffiths, M. D., Rehman, I. U., & Minhas, R. (2021). Fear of COVID-19 and workplace phobia among Pakistani doctors: A survey study. BMC Public Health, 21, 833.

Mahmoudi, H., Saffari, M., Movahedi, M., Sanaeinasab, H., Rashidi-Jahan, H., Pourgholami, M., Poorebrahim, A., Barshan, J., Ghiami, M., Khoshmanesh, S., Potenza, M. N., Lin, C.-Y.*, Pakpour, A. H.* (2021). The mediating role of mental health in associations between COVID-19-related self-stigma, PTSD, quality of life and insomnia among patients recovered from COVID-19. Brain and Behavior, 11, e02138.

Chen, C.-Y., Chen, I.-H., Hou, W.-L., Potenza, M. N., O'Brien, K. S., Lin, C.-Y., & Latner, J. D. (2021). The relationship between children’s problematic Internet-related behaviors and psychological distress during the onset of the COVID-19 pandemic: A longitudinal study. Journal of Addiction Medicine.

Chen, C.-Y., Chen, I.-H., Pakpour, A. H., Lin, C.-Y., & Griffiths, M. D. (2021). Internet-related behaviors and psychological distress among schoolchildren during the COVID-19 school hiatus. Cyberpsychology, Behavior, and Social Networking.

Chen, I.-H., Chen, C.-Y., Pakpour, A. H., Griffiths, M. D., Lin, C.-Y., Li, X.-D., Tsang, H. W. H. (2021). Problematic internet-related behaviors mediate the associations between levels of internet engagement and distress among schoolchildren during COVID-19 lockdown: A longitudinal structural equation modeling study. Journal of Behavioral Addictions, 10(1), 135-148.

Pramukti, I., Strong, C., Sitthimongkol, Y., Setiawan, A., Pandin M. G. R., Yen, C.-F., Lin, C.-Y., Griffiths, M. D., Ko, N.-Y. (2020). Anxiety and suicidal thoughts during the COVID-19 pandemic: A cross-country comparison among Indonesian, Taiwanese, and Thai university students. Journal of Medical Internet Research, 22(12), e24487.

Chen, C.-Y., Chen, I.-H., O'Brien, K. S., Latner, J. D., & Lin, C.-Y. (2021). Psychological distress and internet-related behaviors between schoolchildren with and without overweight during the COVID-19 outbreak. International Journal of Obesity, 45(3), 677-686.

Ahorsu, D. K., Lin, C.-Y., Pakpour, A. H. (2020). The association between health status and insomnia, mental health, and preventive behaviours: The mediating role of fear of COVID-19. Gerontology and Geriatric Medicine, 6, 1-9.

Chang, K.-C., Strong, C., Pakpour, A. H., Griffiths, M. D., & Lin, C.-Y. (2020). Factors related to preventive COVID-19 infection behaviors among people with mental illness. Journal of the Formosan Medical Association, 119(12), 1772-1780.

Lin, C.-Y., Broström, A., Griffiths, M. D., & Pakpour, A. H. (2020). Investigating mediated effects of fear of COVID-19 and COVID-19 misunderstanding in the association between problematic social media use and distress/insomnia. Internet Interventions, 21, 100345.

Chen, I.-H., Chen, C.-Y., Pakpour, A. H., Griffiths, M. D., & Lin, C.-Y. (2020). Internet-related behaviors and psychological distress among schoolchildren during COVID-19 school suspension. Journal of the American Academy of Child and Adolescent Psychiatry, 159(10), 1099-1102.

Lin, C.-Y. (2020). Social reaction toward the 2019 Novel Coronavirus (COVID-19). Social Health & Behavior, 3, 1-2.

3. The authors said "For this study, we used the Arabic version of FCV-19S, which was validated in a previous study.(31)". However, they also provide a section to describe the translation procedure. This is somewhat confusing because I cannot know whether the authors translated the FCV-19S by themselves or the translation descriptions are derived from the reference #31.

4. In ethics, please provide the registered number of the IRB.

5. If possible, I would like to know if there is any demographic difference between the analyzed sample (n=3989) and those who gave a consent but not used for data analysis (n=441). This can provide information to the readers regarding the sample representativeness.

6. In Discussion, the authors should mention that fear may be a factor impacts an individual's protective behaviors, including vaccination uptake using the references I provided below. Therefore, one should have a certain level of fear but not an overwhelming fear.

Wang, P.-W., Ahorsu, D. K., Lin, C.-Y., Chen, I.-H., Yen, C.-F., Kuo, Y.-J., Griffiths, M. D., & Pakpour, A. H. (2021). Motivation to Have COVID-19 Vaccination Explained Using an Extended Protection Motivation Theory Among University Students in China: The Role of Information Sources. Vaccines, 9, 380.

7. Following my previous comment, the authors can make some comments regarding the low willingness of vaccination uptake in some countries (Kukreti et al., 2021), and thus monitoring fear is important.

Kukreti, S., Lu, M.-L., Lin, Y.-H., Strong, C., Lin, C.-Y., Ko, N.-Y., Chen, P.-L., & Ko, W.-C. (2021). Willingness of Taiwan’s healthcare workers and outpatients to vaccinate against COVID-19 during a period without community outbreaks. Vaccines, 9(3), 246.

8. I would encourage the authors reformatted their tables as they are hard to read. Please consult some published articles in PLOS ONE to revise these tables.

9. The references are not in good shape. Specifically, the authors used "Organization WH" to indicate "World Health Organization", which is incorrect. Also, there are some repeated references in the list (e.g., #16 and #19 are the same reference). The authors should check their reference list carefully.

6. PLOS authors have the option to publish the peer review history of their article (what does this mean?). If published, this will include your full peer review and any attached files.

Reviewer #1: **Yes: **Adib Essali

Reviewer #2: No

---

## [Author Response · Author response to Decision Letter 0]

14 Jun 2021

Response letter

 We would like to thank both the editor and reviewers for spending their valuable time reviewing this manuscript. We have thoroughly assessed the comments and implemented them into our revised manuscript, now that this article has been made suitable, we aspire to have this article published in your journal. 

The following letter will report the responses to the reviewers’ comments.

Editor and Reviewer comments:

Author response: We have revised the style requirements and implemented them within our manuscript.

2. We note that your Supporting Information file may include questionnaire items that may have been previously published. The reproduction of previously published work has implications for the copyright that may apply to these publications. We would be grateful if you could clarify whether you have obtained permission from the original copyright holder to republish these items under a CC BY license. If you have not obtained permission to publish these items please remove them from your manuscript. You may wish to replace the text you have removed with relevant question numbers/ brief descriptions of each item; please be sure to include any relevant references and in-text citations.

Author response: We will remove the English version of the FCV-19S from the supporting files. We mentioned in the methods clearly that we used the English version and translated it ourselves to Arabic. We apologize for accidently mentioning “For this study, we used the Arabic version of FCV-19S, which was validated in a previous study.”, we actually translated the questionnaire ourselves and validated it statistically.

3. Please amend the manuscript submission data (via Edit Submission) to include author Salma Khadem alsrouji.

Author response: We have made the necessary amendment.

Author response: We agree and have done as requested above. 

5.Thank you for submitting the above manuscript to PLOS ONE. During our internal evaluation of the manuscript, we found significant text overlap between your submission and the following previously published works.

- https://www.statnews.com/2020/03/16/coronavirus-serious-threat-prepare-not-overreact/

We would like to make you aware that copying extracts from previous publications, especially outside the methods section, word-for-word is unacceptable, even for works which you authored. In addition, the reproduction of text from published reports has implications for the copyright that may apply to the publications.

Please revise the manuscript to rephrase the duplicated text, cite your sources, and provide details as to how the current manuscript advances on previous work. Please note that further consideration is dependent on the submission of a manuscript that addresses these concerns about the overlap in text with published work.

 Author response: We apologise for this, we were not aware of this matter and we don’t remember seeing the above article, which was slightly confusing for us. We have revised the manuscript and have rephrased some sentences. If there are still sentences that show plagiarism, we would appreciate if you could identify them for us and we would be happy to rephrase them. Again you have our deepest apologies.

Reviewers' comments:

Reviewer's Responses to Questions

Comments to the Author

1. Is the manuscript technically sound, and do the data support the conclusions?

Reviewer #1: Partly

Reviewer #2: Partly

2. Has the statistical analysis been performed appropriately and rigorously?

Reviewer #1: Yes

Reviewer #2: Yes

3. Have the authors made all data underlying the findings in their manuscript fully available?

Reviewer #1: Yes

Reviewer #2: Yes

4. Is the manuscript presented in an intelligible fashion and written in standard English?

Reviewer #1: Yes

Reviewer #2: Yes

5. Review Comments to the Author

Reviewer #1: This is a well reported study. However, the following points need clarification:

Author response: We thank the reviewer for their compliment and will address the comments provided below. 

1) The authors state that sample size was calculated for a population of 17, 500, 657 people (line 100). It is well known that much less Syrians were living in Syria in 2020.

Author response: We agree with this, however, there are no credible national data regarding this matter. We took this data from a fact sheet published by the World Health Organization (https://gco.iarc.fr/today/data/factsheets/populations/760-syrian-arab-republic-fact-sheets.pdf). This citation has been added to the manuscript

2) The authors imply that they tested a sample that is representing all Syrians. It is not clear how the 5000 participants were chosen to be invited to take part in the study.

Author response: We apologise for not mentioning the type of sampling. We used a convenience sampling method in the study. This has been added to the methods section. We mentioned in the limitations that credible published national data regarding the socio-demographic variables of Syrians are not available to evaluate the representativeness of our sample.

3) This is web-based cross-sectional study employing social media. It is unlikely that all the people appearing in a random sample would have an online access to social media.

Author response: We apologise for not mentioning the type of sampling. We used a convenience sampling method in the study. This has been added to the methods section.

4) There is a contradiction between the statement (The original FCV-19S was translated into Arabic using a forward-backward translation technique – line 110) and the statement (For this study, we used the Arabic version of FCV-19S, which was validated in a previous study – line 123).

Author response: We apologise the mistake, we did not use the validated Arabic version of FCV-19S. This has been removed from the text.

5) The title “Fear among Syrians: a Proposed Cutoff Score and Validation of the Arabic Fear of COVID-19 Scale- A National Survey” implies that that the results of this survey could generalised to the whole Syrian nation. This does not seem to be the case as the authors acknowledge that They has used “convenience sampling” – line 244, and the results cannot be generalized. This study, thus, cannot be described as a national survey in the title and in the text.

Author response: We agree with the reviewer and have removed “A National Survey” from the title.

6) the emphasis in the title on a “proposed cut-off point” that is useless is not appropriate.

Author response: We understand the reviewers comment, we would like to keep the title like this. We could change the title to “Fear of COVID-19 among Syrians” if the reviewer prefers this one.

Reviewer #2: I believe that the present study can add merits to the present understanding of fear toward COVID-19. To the best of my knowledge, although many studies have translated and validated the Fear of COVID-19 Scale (FCV-19S), none have proposed a cutoff for the FCV-19S. Therefore, the major contribution of the present study is the proposed cutoff with using previous standardized fear scales. Moreover, the present study has the strengths of (1) rigorous translation for the FCV-19S, (2) large sample size of 3989, and (3) adequate statistical methods. Therefore, I read this paper with greatest interests. However, several points should be addressed before I recommend publication. Please see my comments below.

Author response: We appreciate the reviewer’s compliments and will thoroughly address the comments provided below.

1. I would like the authors to provide more information regarding the psychometric evidence of the FCV-19S. The following references are recommended to be included in a revision.

Chang, K.-C., Hou, W.-L., Pakpour, A. H., Lin, C.-Y., Griffiths, M. D. (accepted). Psychometric testing of three COVID-19-related scales among people with mental illness. International Journal of Mental Health and Addiction.

Pakpour, A. H., Griffiths, M. D., Chang, K.-C., Chen, Y.-P., Kuo, Y.-J., & Lin, C.-Y. (2020). Assessing the fear of COVID-19 among different populations: A response to Ransing et al. (2020). Brain, Behavior, and Immunity, 89, 524-525.

Andrade, E. F., Pereira, L. J., Oliveira, A., Orlando, D. R., Alves, D., Guilarducci, J. S., & Castelo, P. M. (2020). Perceived fear of COVID-19 infection according to sex, age and occupational risk using the Brazilian version of the Fear of COVID-19 Scale. Death studies, 1–10. Advance online publication. https://doi.org/10.1080/07481187.2020.1809786

Haktanir, A., Seki, T., & Dilmaç, B. (2020). Adaptation and evaluation of Turkish version of the fear of COVID-19 Scale. Death studies, 1–9. Advance online publication. https://doi.org/10.1080/07481187.2020.1773026

Caycho-Rodríguez, T., Valencia, P. D., Vilca, L. W., Cervigni, M., Gallegos, M., Martino, P., Barés, I., Calandra, M., Rey Anacona, C. A., López-Calle, C., Moreta-Herrera, R., Chacón-Andrade, E. R., Lobos-Rivera, M. E., Del Carpio, P., Quintero, Y., Robles, E., Panza Lombardo, M., Gamarra Recalde, O., Buschiazzo Figares, A., White, M., … Burgos Videla, C. (2021). Cross-cultural measurement invariance of the fear of COVID-19 scale in seven Latin American countries. Death studies, 1–15. Advance online publication. https://doi.org/10.1080/07481187.2021.1879318

Soares, F. R., Afonso, R. M., Martins, A. P., Pakpour, A. H., & Rosa, C. P. (2021). The fear of the COVID-19 Scale: validation in the Portuguese general population. Death studies, 1–7. Advance online publication. https://doi.org/10.1080/07481187.2021.1889722

Moreta-Herrera, R., López-Calle, C., Caycho-Rodríguez, T., Cabezas Guerra, C., Gallegos, M., Cervigni, M., Martino, P., Barés, I., & Calandra, M. (2021). Is it possible to find a bifactor structure in the Fear of COVID-19 Scale (FCV-19S)? Psychometric evidence in an Ecuadorian sample. Death studies, 1–11. Advance online publication. https://doi.org/10.1080/07481187.2021.1914240

Author response: We agree with the author and have implanted this within the introduction section.

2. In the second paragraph of the Introduction, the authors should include the following references, which also support the statement that COVID-19 impacts on individuals' physical and mental health.

Fung, X. C. C.#, Siu, A., Potenza, M. N., O'Brien, K. S., Latner, J. D., Chen, C.-Y., Chen, I.-H., & Lin, C.-Y. (2021). Problematic use of internet-related activities and perceived weight stigma in schoolchildren: A longitudinal study across different epidemic periods of COVID-19 in China. Frontiers in Psychiatry.

Malik, S., Ullah, I., Irfan, M., Ahorsu, D. K., Lin, C.-Y., Pakpour, A. H., Griffiths, M. D., Rehman, I. U., & Minhas, R. (2021). Fear of COVID-19 and workplace phobia among Pakistani doctors: A survey study. BMC Public Health, 21, 833.

Mahmoudi, H., Saffari, M., Movahedi, M., Sanaeinasab, H., Rashidi-Jahan, H., Pourgholami, M., Poorebrahim, A., Barshan, J., Ghiami, M., Khoshmanesh, S., Potenza, M. N., Lin, C.-Y.*, Pakpour, A. H.* (2021). The mediating role of mental health in associations between COVID-19-related self-stigma, PTSD, quality of life and insomnia among patients recovered from COVID-19. Brain and Behavior, 11, e02138.

Chen, C.-Y., Chen, I.-H., Hou, W.-L., Potenza, M. N., O'Brien, K. S., Lin, C.-Y., & Latner, J. D. (2021). The relationship between children’s problematic Internet-related behaviors and psychological distress during the onset of the COVID-19 pandemic: A longitudinal study. Journal of Addiction Medicine.

Chen, C.-Y., Chen, I.-H., Pakpour, A. H., Lin, C.-Y., & Griffiths, M. D. (2021). Internet-related behaviors and psychological distress among schoolchildren during the COVID-19 school hiatus. Cyberpsychology, Behavior, and Social Networking.

Chen, I.-H., Chen, C.-Y., Pakpour, A. H., Griffiths, M. D., Lin, C.-Y., Li, X.-D., Tsang, H. W. H. (2021). Problematic internet-related behaviors mediate the associations between levels of internet engagement and distress among schoolchildren during COVID-19 lockdown: A longitudinal structural equation modeling study. Journal of Behavioral Addictions, 10(1), 135-148.

Pramukti, I., Strong, C., Sitthimongkol, Y., Setiawan, A., Pandin M. G. R., Yen, C.-F., Lin, C.-Y., Griffiths, M. D., Ko, N.-Y. (2020). Anxiety and suicidal thoughts during the COVID-19 pandemic: A cross-country comparison among Indonesian, Taiwanese, and Thai university students. Journal of Medical Internet Research, 22(12), e24487.

Chen, C.-Y., Chen, I.-H., O'Brien, K. S., Latner, J. D., & Lin, C.-Y. (2021). Psychological distress and internet-related behaviors between schoolchildren with and without overweight during the COVID-19 outbreak. International Journal of Obesity, 45(3), 677-686.

Ahorsu, D. K., Lin, C.-Y., Pakpour, A. H. (2020). The association between health status and insomnia, mental health, and preventive behaviours: The mediating role of fear of COVID-19. Gerontology and Geriatric Medicine, 6, 1-9.

Chang, K.-C., Strong, C., Pakpour, A. H., Griffiths, M. D., & Lin, C.-Y. (2020). Factors related to preventive COVID-19 infection behaviors among people with mental illness. Journal of the Formosan Medical Association, 119(12), 1772-1780.

Lin, C.-Y., Broström, A., Griffiths, M. D., & Pakpour, A. H. (2020). Investigating mediated effects of fear of COVID-19 and COVID-19 misunderstanding in the association between problematic social media use and distress/insomnia. Internet Interventions, 21, 100345.

Chen, I.-H., Chen, C.-Y., Pakpour, A. H., Griffiths, M. D., & Lin, C.-Y. (2020). Internet-related behaviors and psychological distress among schoolchildren during COVID-19 school suspension. Journal of the American Academy of Child and Adolescent Psychiatry, 159(10), 1099-1102.

Lin, C.-Y. (2020). Social reaction toward the 2019 Novel Coronavirus (COVID-19). Social Health & Behavior, 3, 1-2.

Author response: Thank you for providing the citations to further support our statement within the manuscript. These citations have been implemented within the manuscript.

3. The authors said "For this study, we used the Arabic version of FCV-19S, which was validated in a previous study.(31)". However, they also provide a section to describe the translation procedure. This is somewhat confusing because I cannot know whether the authors translated the FCV-19S by themselves or the translation descriptions are derived from the reference #31.

Author response: We apologise the mistake, we did not use the validated Arabic version of FCV-19S. This has been removed from the text.

4. In ethics, please provide the registered number of the IRB.

Author response: The IRB ant SPU ethically approved our study but did grant us a registered number.

5. If possible, I would like to know if there is any demographic difference between the analyzed sample (n=3989) and those who gave a consent but not used for data analysis (n=441). This can provide information to the readers regarding the sample representativeness.

Author response: The 441 participants who did not meet the inclusion criteria were either below 18, residing outside Syria, or uncompleted survey. This is the information we have about those excluded. We added asentence in the results section.

6. In Discussion, the authors should mention that fear may be a factor impacts an individual's protective behaviors, including vaccination uptake using the references I provided below. Therefore, one should have a certain level of fear but not an overwhelming fear.

Wang, P.-W., Ahorsu, D. K., Lin, C.-Y., Chen, I.-H., Yen, C.-F., Kuo, Y.-J., Griffiths, M. D., & Pakpour, A. H. (2021). Motivation to Have COVID-19 Vaccination Explained Using an Extended Protection Motivation Theory Among University Students in China: The Role of Information Sources. Vaccines, 9, 380.

Author response: A very nice point suggested by the reviewer, we have implemented this within the discussion

7. Following my previous comment, the authors can make some comments regarding the low willingness of vaccination uptake in some countries (Kukreti et al., 2021), and thus monitoring fear is important.

Kukreti, S., Lu, M.-L., Lin, Y.-H., Strong, C., Lin, C.-Y., Ko, N.-Y., Chen, P.-L., & Ko, W.-C. (2021). Willingness of Taiwan’s healthcare workers and outpatients to vaccinate against COVID-19 during a period without community outbreaks. Vaccines, 9(3), 246.

Author response: We understand the suggesting provided by the reviewer; however, we would like to avoid deviating from the topic of the paper. If the reviewer insists on adding this point we would be happy to do so.

8. I would encourage the authors reformatted their tables as they are hard to read. Please consult some published articles in PLOS ONE to revise these tables.

Author response: We have read the table formatting information and had a look at a few published studies by the PLOS ONE journal. We hope the tables are in the correct format now.

9. The references are not in good shape. Specifically, the authors used "Organization WH" to indicate "World Health Organization", which is incorrect. Also, there are some repeated references in the list (e.g., #16 and #19 are the same reference). The authors should check their reference list carefully.

Author response: We apologize for this, and have made the necessary amendments.

---

## [Decision Letter · Decision Letter 1]

7 Jul 2021

PONE-D-21-14780R1

Fear among Syrians: a Proposed Cutoff Score and Validation of the Arabic Fear of COVID-19 Scale

PLOS ONE

Dear Dr. Mohsen,

Thank you for submitting your manuscript to PLOS ONE. After careful consideration, we feel that it has merit but does not fully meet PLOS ONE’s publication criteria as it currently stands. Therefore, we invite you to submit a revised version of the manuscript that addresses the points raised during the review process.

We look forward to receiving your revised manuscript.

Kind regards,

Ali B. Mahmoud, Ph.D.

Academic Editor

PLOS ONE

Journal Requirements:

Reviewers' comments:

Reviewer's Responses to Questions

**Comments to the Author**

1. If the authors have adequately addressed your comments raised in a previous round of review and you feel that this manuscript is now acceptable for publication, you may indicate that here to bypass the “Comments to the Author” section, enter your conflict of interest statement in the “Confidential to Editor” section, and submit your "Accept" recommendation.

Reviewer #1: (No Response)

Reviewer #2: All comments have been addressed

2. Is the manuscript technically sound, and do the data support the conclusions?

Reviewer #1: Partly

Reviewer #2: Yes

3. Has the statistical analysis been performed appropriately and rigorously? 

Reviewer #1: Yes

Reviewer #2: Yes

4. Have the authors made all data underlying the findings in their manuscript fully available?

Reviewer #1: Yes

Reviewer #2: Yes

5. Is the manuscript presented in an intelligible fashion and written in standard English?

Reviewer #1: Yes

Reviewer #2: Yes

6. Review Comments to the Author

Reviewer #1: The authors have responded to some but not all comments. The calculation of sample size is pointless and could be misleading. The authors agree that the population of Syria at the time of the study was unknown, i.e. it is not possible to calculate the sample size. In addition, sample size is calculated in order to generalise the results, and the results of this study cannot be generalised. I recommend removing the section about sample size.

My comment about the cutoff point stands. A 50/50 cutoff point is pointless. It is mere chance and the questionnaire is adding nothing.

Reviewer #2: The authors have improved their work after considering the comments from both reviewers. However, some minor works are needed further. Specifically, the authors did not reformat their reference list in a good shape. The Organization WH still shows up in the reference list. Please make sure the correctness of the reference list.

7. PLOS authors have the option to publish the peer review history of their article (what does this mean?). If published, this will include your full peer review and any attached files.

Reviewer #1: **Yes: **Adib Essali

Reviewer #2: No

---

## [Author Response · Author response to Decision Letter 1]

7 Jul 2021

Response letter

 We would like to thank both the editor and reviewers for spending their valuable time reviewing this manuscript. We have thoroughly assessed the comments and implemented them into our revised manuscript, now that this article has been made suitable, we aspire to have this article published in your journal. 

The following letter will report the responses to the reviewers’ comments.

ournal Requirements:

Author response: We apologize and have edited the references mentioned by the reviewer below. W are not aware of citing papers that have been retracted within the manuscript. If there is any we would appreciate highlighting any out for us to manage appropriately.

Reviewers' comments:

Reviewer's Responses to Questions

Comments to the Author

1. If the authors have adequately addressed your comments raised in a previous round of review and you feel that this manuscript is now acceptable for publication, you may indicate that here to bypass the “Comments to the Author” section, enter your conflict of interest statement in the “Confidential to Editor” section, and submit your "Accept" recommendation.

Reviewer #1: (No Response)

Reviewer #2: All comments have been addressed

2. Is the manuscript technically sound, and do the data support the conclusions?

Reviewer #1: Partly

Reviewer #2: Yes

3. Has the statistical analysis been performed appropriately and rigorously?

Reviewer #1: Yes

Reviewer #2: Yes

4. Have the authors made all data underlying the findings in their manuscript fully available?

Reviewer #1: Yes

Reviewer #2: Yes

5. Is the manuscript presented in an intelligible fashion and written in standard English?

Reviewer #1: Yes

Reviewer #2: Yes

6. Review Comments to the Author

Reviewer #1: The authors have responded to some but not all comments. The calculation of sample size is pointless and could be misleading. The authors agree that the population of Syria at the time of the study was unknown, i.e. it is not possible to calculate the sample size. In addition, sample size is calculated in order to generalise the results, and the results of this study cannot be generalised. I recommend removing the section about sample size.

My comment about the cutoff point stands. A 50/50 cutoff point is pointless. It is mere chance and the questionnaire is adding nothing.

Author response: We removed the section on calculating sample size.

Reviewer #2: The authors have improved their work after considering the comments from both reviewers. However, some minor works are needed further. Specifically, the authors did not reformat their reference list in a good shape. The Organization WH still shows up in the reference list. Please make sure the correctness of the reference list.

Author response: We have changed the reference of Organization WH to WHO.

---

## [Decision Letter · Decision Letter 2]

11 Jul 2021

PONE-D-21-14780R2

Fear among Syrians: a Proposed Cutoff Score and Validation of the Arabic Fear of COVID-19 Scale

PLOS ONE

Dear Dr. Mohsen,

The reviewers have now recommended publication. However, before proceeding with accepting your paper for publication in PLOS ONE, I identify a few minor improvements that would benefit the quality of your research. Therefore, I invite you to submit a revised version of the manuscript that addresses the following points alongside a response file showing how you have responded to each of my comments.

Please report the average variance extracted (AVE) scores and items’ outer loadings as an assessment of construct validity for all of the measures used in the study. Also, computing the composite reliability (CR) would be great.The script does not describe how the discriminant validity was assessed (e.g., HTMT). Please, revise accordingly.Given the ongoing turmoil in Syria, literature on “Wartime Crisis Perceptions” should be brought up in the discussion, mainly concerning not considering this variable in the study, hence offering directions for future research. Similarly, COVID-19 perception (defined as “as the perceived probability of discomfort and/or worry, during COVID-19 pandemic, concerning the pandemic adverse health, economic and social ramifications articulated as disruptions to the people’s pre-pandemic everyday life – lead to redefining of the everyday life to the new normal.”) which conceptually should occur before developing fears and other negative experiences is warranted a discussion here too.Ensure the accuracy of terms used in the paper (e.g., anxiety, fear are emotions that our perception of events would trigger).The whole manuscript would benefit from proper proofreading.I found some similarity issues that you will need to remedy. Please, see attached the similarity check report and paraphrase/cite accordingly.Finally, given the study context, an Arabic abstract of the research, as supplementary material, would add more visibility to your paper. Also, ensure a list of the translated items is offered in Arabic as supplementary material too.

We look forward to receiving your revised manuscript.

Kind regards,

Ali B. Mahmoud, Ph.D.

Academic Editor

PLOS ONE

Journal Requirements:

Reviewers' comments:

Reviewer's Responses to Questions

**Comments to the Author**

1. If the authors have adequately addressed your comments raised in a previous round of review and you feel that this manuscript is now acceptable for publication, you may indicate that here to bypass the “Comments to the Author” section, enter your conflict of interest statement in the “Confidential to Editor” section, and submit your "Accept" recommendation.

Reviewer #1: All comments have been addressed

Reviewer #2: (No Response)

2. Is the manuscript technically sound, and do the data support the conclusions?

Reviewer #1: (No Response)

Reviewer #2: Yes

3. Has the statistical analysis been performed appropriately and rigorously? 

Reviewer #1: (No Response)

Reviewer #2: Yes

4. Have the authors made all data underlying the findings in their manuscript fully available?

Reviewer #1: (No Response)

Reviewer #2: Yes

5. Is the manuscript presented in an intelligible fashion and written in standard English?

Reviewer #1: (No Response)

Reviewer #2: Yes

6. Review Comments to the Author

Reviewer #1: Thank you for making the changes. I hope that this will be the beginning of a successful research career. Please remember that panning and written a protocol BEFORE conducting research in the most important point in this field. Seek advice and discuss your protocol with many people before conducting studies in the future. With my best wishes.

Reviewer #2: The authors have revised the manuscript again and I have no more comments now. I am satisfied with the current version.

7. PLOS authors have the option to publish the peer review history of their article (what does this mean?). If published, this will include your full peer review and any attached files.

Reviewer #1: **Yes: **Adib Essali

Reviewer #2: No

---

## [Author Response · Author response to Decision Letter 2]

22 Sep 2021

Response letter

 We would like to thank both the editor and reviewers for spending their valuable time reviewing this manuscript. We have thoroughly assessed the comments and implemented them into our revised manuscript, now that this article has been made suitable, we aspire to have this article published in your journal. 

The following letter will report the responses to the reviewers’ comments.

We've checked your submission and before we can proceed, we need you to address the following issues:

Author response: We will address the following issue provided.

2.Please include a separate legend for each figure in your manuscript.

Author response: We have included figure captions in the manuscript text in read order, immediately following the paragraph where the figure is first cited.

---

## [Editor Report · Decision Letter 3]

24 Sep 2021

PONE-D-21-14780R3Fear among Syrians: a Proposed Cutoff Score for the Arabic Fear of COVID-19 ScalePLOS ONE

Dear Dr. Mohsen,

Thank you for submitting your manuscript to PLOS ONE. After careful consideration, I feel that it has merit but does not fully meet PLOS ONE’s publication criteria as it currently stands. Therefore, I invite you to submit a revised version of the manuscript that addresses,** IN FULL**, the points raised during the previous round of the peer review. Please highlight or track **all of the changes** made in your revision in response to this decision letter. In addition, please, include a** table/response file **(This requirement is already mentioned below) detailing how each of the comments has been addressed in your revised manuscript. This is very critical for further consideration. For your convenience, I list the comments again herewith.

*Please report the average variance extracted (AVE) scores and items’ outer loadings as an assessment of construct validity for all of the measures used in the study. Also, computing the composite reliability (CR) would be great.*

*The script does not describe how the discriminant validity was assessed (e.g., HTMT). Please, revise accordingly.*

*Given the ongoing turmoil in Syria, literature on “Wartime Crisis Perceptions” or “Personal Experience of Wartime Crisis (PEoWTC)” should be defined with proper citation(s) and brought up in the discussion, mainly concerning not considering this variable in the study hence offering directions for future research. Similarly, COVID-19 perception (defined as “as the perceived probability of discomfort and/or worry, during COVID-19 pandemic, concerning the pandemic adverse health, economic and social ramifications articulated as disruptions to the people’s pre-pandemic everyday life – lead to redefining of the everyday life to the new normal.”) which conceptually should occur before developing fears and other negative experiences is warranted a discussion here too.*

*Ensure the accuracy of terms used in the paper (e.g., anxiety, fear are emotions that our perception of events would trigger).*

*The whole manuscript would benefit from proper proofreading.*

*I found some similarity issues that you will need to remedy. Please, see attached the similarity check report and paraphrase/cite accordingly.*

*Finally, given the study context, an Arabic abstract of the research, as supplementary material, would add more visibility to your paper. Also, ensure a list of the translated items is offered in Arabic as supplementary material too.*

We look forward to receiving your revised manuscript.

Kind regards,

Ali B. Mahmoud, Ph.D.

Academic Editor

PLOS ONE
---

## [Author Response · Author response to Decision Letter 3]

5 Feb 2022

Response letter

 We would like to thank both the editor and reviewers for spending their valuable time reviewing this manuscript. We have thoroughly assessed the comments and implemented them into our revised manuscript, now that this article has been made suitable, we aspire to have this article published in your journal. 

The following letter will report the responses to the reviewers’ comments.

1. Please report the average variance extracted (AVE) scores and items’ outer loadings as an assessment of construct validity for all of the measures used in the study. Also, computing the composite reliability (CR) would be great.

Author response: We have added a table that includes factor loading, corrected item total correlation, cronbach’s alpha, inter item correlation range, skewness, kurtosis, and Kaiser meyer-olkin for each item of the FCV-19S. The data is presented in table 3. We also calculated the AVE and CR and reported them in the results section.

2. The script does not describe how the discriminant validity was assessed (e.g., HTMT). Please, revise accordingly.

Author response: We have reported the discriminant validity.

3. Given the ongoing turmoil in Syria, literature on “Wartime Crisis Perceptions” should be brought up in the discussion, mainly concerning not considering this variable in the study, hence offering directions for future research. Similarly, COVID-19 perception (defined as “as the perceived probability of discomfort and/or worry, during COVID-19 pandemic, concerning the pandemic adverse health, economic and social ramifications articulated as disruptions to the people’s pre-pandemic everyday life – lead to redefining of the everyday life to the new normal.”) which conceptually should occur before developing fears and other negative experiences is warranted a discussion here too.

Author response: A very important point, we have included this point in the limitations section. Highlighting the need for this point to be further assessed.

4. Ensure the accuracy of terms used in the paper (e.g., anxiety, fear are emotions that our perception of events would trigger).

Author response: We have revised the manuscript for the point mentioned above.

5. The whole manuscript would benefit from proper proofreading.

Author response: We have revised the manuscript.

6. I found some similarity issues that you will need to remedy. Please, see attached the similarity check report and paraphrase/cite accordingly.

Author response: We have rephrased as many sentences as we could from the file sent. However, there are many phrases and words we cannot change, for example “fear of COVID-19 scale” we noticed these were picked up. Also, places we can not change include the affiliation section. We hope the manuscript is better now.

7. Finally, given the study context, an Arabic abstract of the research, as supplementary material, would add more visibility to your paper. Also, ensure a list of the translated items is offered in Arabic as supplementary material too.

Author response: We have translated the abstract into Arabic and have attached it as a supplementary file. The items have been translated into Arabic and are available in the supplementary material as S3.

---

## [Editor Report · Decision Letter 4]

8 Feb 2022

Fear among Syrians: a Proposed Cutoff Score for the Arabic Fear of COVID-19 Scale

PONE-D-21-14780R4

Dear Dr. Mohsen,

We’re pleased to inform you that your manuscript has been judged scientifically suitable for publication and will be formally accepted for publication once it meets all outstanding technical requirements.

Kind regards,

Ali B. Mahmoud, Ph.D.

Academic Editor

PLOS ONE
---

## [Editor Report · Acceptance letter]

4 Mar 2022

PONE-D-21-14780R4 

Fear among Syrians: a Proposed Cutoff Score for the Arabic Fear of COVID-19 Scale 

Dear Dr. Mohsen:

I'm pleased to inform you that your manuscript has been deemed suitable for publication in PLOS ONE. Congratulations! Your manuscript is now with our production department. 

Kind regards, 

on behalf of

Dr. Ali B. Mahmoud 

Academic Editor

PLOS ONE